# Empagliflozin Is Not Renoprotective in Non-Diabetic Rat Models of Chronic Kidney Disease

**DOI:** 10.3390/biomedicines10102509

**Published:** 2022-10-07

**Authors:** Silvie Hojná, Zoe Kotsaridou, Zdeňka Vaňourková, Hana Rauchová, Michal Behuliak, Petr Kujal, Michaela Kadlecová, Josef Zicha, Ivana Vaněčková

**Affiliations:** 1Institute of Physiology, Czech Academy of Sciences, 14220 Prague, Czech Republic; 2Department of Biotechnology, Agricultural University, 11855 Athens, Greece; 3Institute for Clinical and Experimental Medicine, 14220 Prague, Czech Republic; 4Department of Pathology, Third Faculty of Medicine, Charles University, 14220 Prague, Czech Republic

**Keywords:** SGLT-2 inhibition, proteinuria, uninephrectomized salt-loaded, two-kidney, one-clip hypertension, fawn-hooded hypertensive rat

## Abstract

Gliflozins (sodium-glucose transporter-2 inhibitors) exhibited renoprotective effects not only in diabetic but also in non-diabetic patients with chronic kidney disease (CKD). Controversial results were reported in experimental non-diabetic models of CKD. Therefore, we examined empagliflozin effects in three CKD models, namely, in fawn-hooded hypertensive (FHH) rats, uninephrectomized salt-loaded (UNX + HS) rats, and in rats with Goldblatt hypertension (two-kidney, one-clip 2K1C) that were either untreated or treated with empagliflozin (10 mg/kg/day) for eight weeks. Plethysmography blood pressure (BP) was recorded weekly, and renal parameters (proteinuria, plasma urea, creatinine clearance, and sodium excretion) were analyzed three times during the experiment. At the end of the study, blood pressure was also measured directly. Markers of oxidative stress (TBARS) and inflammation (MCP-1) were analyzed in kidney and plasma, respectively. Body weight and visceral adiposity were reduced by empagliflozin in FHH rats, without a significant effect on BP. Experimentally induced CKD (UNX + HS and 2K1C) was associated with a substantial increase in BP and relative heart and kidney weights. Empagliflozin influenced neither visceral adiposity nor BP in these two models. Although empagliflozin increased sodium excretion, suggesting effective SGLT-2 inhibition, it did not affect diuresis in any experimental model. Unexpectedly, empagliflozin did not provide renoprotection because proteinuria, plasma urea, and plasma creatinine were not lowered by empagliflozin treatment in all three CKD models. In line with these results, empagliflozin treatment did not decrease TBARS or MCP-1 levels in either model. In conclusion, empagliflozin did not provide the expected beneficial effects on kidney function in experimental models of CKD.

## 1. Introduction

The inhibition of the sodium-glucose transporter-2 (SGLT-2) by gliflozins promotes glucose and sodium excretion. Positive renoprotective effects were demonstrated not only in diabetic patients who were treated with gliflozins in addition to a standard antidiabetic therapy [1,2] but also in non-diabetic patients [3]. The CREDENCE trial [4] demonstrated the renoprotective effects of canagliflozin in diabetic patients, while the DAPA-CKD trial [5] also showed the efficacy of dapagliflozin in non-diabetic patients with chronic kidney disease. Moreover, the DAPA-HF and EMPEROR-reduced trials also showed improved cardiovascular outcomes in heart failure patients [6]. Several mechanisms were suggested for their beneficial effects, from hemodynamic and natriuretic effects (the reduction in BP; to the modulation of tubulo-glomerular feedback, and the reduction in sympathetic hyperactivity); to the metabolic factors due to their glycosuric effects (the improvement of insulin sensitivity and the lowering of plasma triglycerides and uric acid); and to the attenuation of oxidative stress, inflammation, and fibrosis. The beneficial renoprotective effects of different gliflozins were demonstrated in various diabetic and prediabetic rat and mice models [7,8,9,10,11]. However, only limited and sometimes contradictory information is available in non-diabetic animals with chronic kidney disease. While Zhang et al. [12], Rajasekaran et al. [13], and Li et al. [14] found no renoprotective effects of gliflozins in rats after 5/6 nephrectomy, Kim et al. [15] and Wan et al. [16] reported on partial benefits in uninephrectomized rats, and Zeng et al. [17] demonstrated renoprotective effects due to reduced fibrosis in the same model. The anti-fibrotic effects of gliflozins in kidneys were also found in the ureteral obstruction model [18] and the angiotensin II-induced model [19]. In the latter model, the additive renoprotective effects of empagliflozin treatment combined with losartan were shown by Reyes-Pardo [20]. Moreover, Ali et al. [18] reported on the amelioration of adenine-induced CKD with canagliflozin due to a reduction in renal inflammation and oxidative stress. 

Our previous studies in three non-diabetic hypertensive models (hypertensive Ren-2 transgenic rats, hereditary hypertriglyceridemic rats, and spontaneously hypertensive rats expressing the human CRP gene) [21,22,23] demonstrated the beneficial effects of SGLT-2 inhibition without a glucose-lowering effect. In all these models, empagliflozin treatment led to a reduction in body weight and visceral adiposity, while the additional effects depended on the model used—an improvement of metabolic parameters and hepatic metabolism (hHTG), a reduction in blood pressure (TGR), and the attenuation of oxidative stress and inflammation in kidneys (SHR-CRP). 

Three different models of chronic kidney disease with different pathophysiological backgrounds were selected for the present study—a genetic model of hypertension-associated renal failure, namely, a fawn-hooded hypertensive (FHH) rat, in which hypertension, proteinuria, and focal glomerulosclerosis already develops at young age [24]. Moreover, two models of kidney deterioration induced by experimental procedures were used in this study: either a uninephrectomy combined with increased sodium intake (UNX + HS) or a stenosis of renal artery–two-kidney, one-clip (2K1C) Goldblatt hypertension [25].

We hypothesized that empagliflozin would have renoprotective effects in these three models of chronic kidney disease with different pathological backgrounds, i.e., in rats genetically predisposed to renal impairment (FHH) and in two experimentally induced CKD models (UNX + HS or 2K1C).

## 2. Materials and Methods

### 2.1. Animals

Animals were housed at 23o C under a 12 h light/dark cycle, fed maintenance diet for rats and mice (Altromin-1320, Lage, Germany, 11% fat, 24% protein, 65% carbohydrates, 0.45 % NaCl), and given tap water ad libitum. Empagliflozin (Jardiance, Boehringer, Ingelheim, Germany), at a dose of 10 mg/kg/day, was mixed into the Altromin diet (so that the composition of both diets was the same) and given for 8 weeks. The amount of empagliflozin added to the diet was calculated according to our previous studies [21,22,23]. All procedures and experimental protocols were approved by the Ethical Committee of the Institute of Physiology, Czech Academy of Sciences (Protocol Nr. 47/2019), and conformed to the European Convention on Animal Protection and Guidelines on Research Animal Use (Directive 2010/63/EU). Three different models of chronic kidney disease were used within the study.

#### 2.1.1. Fawn-Hooded Hypertensive Rats

Male fawn-hooded hypertensive rats were treated with empagliflozin from the age of 12 weeks, for eight weeks.

#### 2.1.2. Two-Kidney One-Clip (2K1C) Goldblatt Hypertension

Male Wistar rats underwent a sham operation or clipping of the right renal artery (2K1C) at the age of 10 weeks [25]. Eight weeks later, empagliflozin treatment was started and administered for eight weeks.

#### 2.1.3. Uninephrectomized Rats on High-Salt Intake

Male normotensive Hannover Sprague Dawley rats underwent uninephrectomy (UNX) at the age of 7 weeks [25], and control rats were sham operated. One week later, high-salt diet feeding (2% NaCl) and empagliflozin treatment were started. This regimen was applied for eight weeks (Figure 1).

Body weight was monitored throughout the whole experiment and measured once a week. Systolic blood pressure was measured weekly during the experiment using tail plethysmography (HatterasInstruments, Cary, NC, USA). Food consumption was monitored throughout the experiment.

### 2.2. Biochemical Analysis

Three times during the experiment (at weeks 0, 4, and 7), the animals were placed in individual metabolic cages for the evaluation of kidney function using 24-h urine collection. Urinary proteinuria, sodium, and glucose were determined. Urinary protein was measured using the Folin method, with bovine serum albumin as a standard [26]. Plasma creatinine was measured by the FUJI DRI-CHEM analyzer using appropriate slides for creatinine CRE-P III (Fujifilm Corp, Tokyo, Japan). Plasma urea were determined using kit (Erba Lachema, Brno, Czech Republic). Lipoperoxidation products were assessed based on levels of thiobarbituric acid-reactive substances (TBARS) by assaying the reaction with thiobarbituric acid. MCP-1 levels were determined in urine using ELISA kit (Invitrogen, Carlsbad, CA, USA). At the end of the study, direct BP measurement was performed under light isoflurane anesthesia [27]. Then, the animals were sacrificed and plasma and tissue samples were collected and stored at −80 °C for further biochemical analysis.

### 2.3. Statistical Analysis

All data are expressed as means ± SEM. Statistical analysis was done by one-way analysis of variance (ANOVA) with Bonferoni test, using the Graph-Pad Prism software (Graph Pad Software, San Diego, CA, USA). The differences were considered significant at the *p* < 0.05.

## 3. Results

### 3.1. Effects of Empagliflozin on Body Weight, Weights of Fat Depots, and Blood Pressure

All three rat models gained weight during the study. However, there was decrease in body weight in uninephrectomized Hannover Sprague Dawley (HanSD) rats on high-salt diet (UNX + HS + empa), while renal artery stenosis (2K1C + empa) in Wistar rats had no effect on body weight. Empagliflozin (empa) decreased body weight in hypertensive fawn-hooded rats (FHH), while it had no effect on body weight in uninephrectomized rats on a high-salt diet or in rats with renal artery stenosis (Figure 2A,C,E and Table 1, Table 2 and Table 3). The relative food consumption was not different between untreated and empagliflozin-treated animals (14.3 ± 1.4 vs. 13.8 ± 1.2 in FHH rats; 15.2 ± 1.6 vs. 16.8 ± 2.1 in UNX+HS rats, and 12.8 ± 1.6 vs. 14.3 ± 1.4 in 2K1C rats, NS; g/100 g BW).

There was no change in blood pressure (measured by tail-cuff plethysmography) in FHH rats during the experiment (Table 1, Table 2 and Table 3). Moreover, empagliflozin treatment did not modify blood pressure and relative heart weight in this strain. In contrast, blood pressure substantially increased (by about 60 mm Hg in HanSD-UNX+HS and 40 mm Hg in Wistar-2K1C) following surgical procedures in both experimentally induced CKD models (Figure 2B,D,F). This was followed by an increase in their relative heart weights. Empagliflozin had no effect on tail-cuff BP in either experimental group. This was also confirmed by a direct BP measurement at the end of the study. Visceral adiposity was decreased by empagliflozin treatment only in FHH rats. Relative kidney weight was increased in empagliflozin-treated FHH and in uninephrectomized salt-loaded rats, while it was unchanged in 2-kidney-1-clip rats.

### 3.2. Effects of Empagliflozin on Renal Parameters

Urine production was not changed in FHH during the experiment, but as expected it was substantially decreased following nephrectomy, while renal artery stenosis had no effect on this parameter (Figure 3A,C,E). Empagliflozin treatment did not affect diuresis in either of three examined models of CKD, although sodium excretion was increased in FHH and in uninephrectomized HanSD rats, suggesting an effective SGLT-2 blockade (Figure 3B,D). In Wistar-2K1C, increased sodium excretion was observed with no further effect on empagliflozin administration.

Proteinuria increased in all three CKD models during the study (Figure 4). Unexpectedly, empagliflozin treatment not only did not prevent the increase in proteinuria in CKD models but even enhanced it, with this effect being statistically significant in FHH rats. In line with this finding, the plasma urea level was increased in this particular rat strain at the end of the study. In contrast, similar values of plasma urea were found in HanSD-UNX + HS, with a profound increase in this particular rat strain at week 4 and lower values in empagliflozin-treated 2K1C Wistar rats. 

At the end of the study, plasma creatinine was decreased in HanSD-UNX rats, while it was not changed in Wistar-2K1C and in FHH following empagliflozin treatment (Figure 5A,C,E).

### 3.3. Effects of Empagliflozin on Oxidative Stress and ROS Production

At the end of the study, elevated plasma MCP-1 levels (a marker of advanced CKD) were found in uninephrectomized empagliflozin-treated HanSD, while empagliflozin had no effect in other groups (Figure 5B,D,F). In the kidneys, lipoperoxidation products TBARS were increased only due to uninephrectomy, with no effect of empagliflozin treatment on this parameter in either of the three experimental models (Table 1, Table 2 and Table 3). 

## 4. Discussion

In the present study, empagliflozin did not provide any evidence for renoprotection in three non-diabetic models of chronic kidney disease with a different pathophysiological background. In fact, empagliflozin did not improve proteinuria, plasma urea or plasma creatinine, and it did not decrease oxidative stress or inflammation in either model. This lack of kidney protection in our experimental models of CKD is in sharp contrast to the renoprotective effects of SGLT-2 inhibition demonstrated in several big clinical trials (EMPA-REG, DAPA-CKD) [1,5] in non-diabetic patients. The question arises whether these negative results could be ascribed to the relatively short duration of our studies, the gliflozin type and dose administered, the rat strain used, or a relatively early stage of kidney disease in our models. However, experimental studies of other investigators provide rather conflicting results ranging from beneficial effects [17,18,19,28], through disputable findings [15,16], to completely negative results [12,13,14].

Nevertheless, the pleiotropic effects of gliflozins were shown under non-diabetic conditions in many human and experimental studies, including ours. Our previous studies in several hypertensive non-diabetic models demonstrated that their benefits were mainly due to their effects on metabolic and hemodynamic parameters, with less effect on renal or cardiac parameters. Thus, a substantial blood-pressure lowering effect of empagliflozin treatment, together with decreasing body weight and adiposity, was achieved in Ren-2 transgenic rats, a model of angiotensin II-dependent hypertension, but there was little effect on renal parameters [22] and no effect on cardiac function (unpublished results). Similarly, we did not demonstrate the improvement of cardiac function in hereditary hypertriglyceridemic rats, in which beneficial effects were mediated mainly through the improvement of hepatic lipid metabolism [21]. In contrast, in SHR-CRP rats, empagliflozin benefit was based on reduced renal inflammation and oxidative stress [23], which is in line with the findings of Ali et al. [18], who also demonstrated a reduction in renal inflammation and ROS production in adenine-induced CKD. In addition, several studies ascribed renoprotection to the anti-fibrotic effects of gliflozins [17,19,28].

Based on the existing experimental studies, no general conclusion could be reached, suggesting that better results could be expected if the treatment would be prolonged. The problem is that there are no experimental studies evaluating gliflozin effects in the long-term setting, i.e., for months instead of weeks. Thus, the longest study, administering dapagliflozin for 12 weeks [12] or TA-1887 for 10 weeks [14], did not produce positive results, while 10-week treatment with a very low dose of empagliflozin [17] was renoprotective. In humans or in experimental animals with focal segmental glomerulosclerosis [13], eight-week dapagliflozin treatment did not modify renal hemodynamics or did not attenuate proteinuria. In contrast, one- or two-week treatments with empagliflozin or luseogliflozin demonstrated both beneficial and unfavorable effects [16,19,28]. 

It is also difficult to evaluate the sensitivity of the rat strain to gliflozin treatment. However, it seems that studies with Sprague Dawley (SD) rats demonstrated more negative results than those performed on Wistar rats. In fact, there is only one study demonstrating the beneficial effect of gliflozin in SD rats [19] compared to four negative studies [12,13,14,15], while the opposite is true for Wistar rats [16,17,18,20,28]. This might be due to the higher sensitivity of SD rats to the development of proteinuric kidney disease following nephrectomy (Kala et al., unpublished results), which could be further aggravated by a high-salt diet. 

Gliflozin type and dose used should also be taken into consideration. Empagliflozin seems to be used more frequently than dapagliflozin or canagliflozin, or a newly synthetized luseogliflozin. The dose of empagliflozin used in experimental studies is usually 10 mg/kg/day (similar to humans), which, however, yielded inconsistent results. Even a very low dose of empagliflozin (0.6 mg/kg/day) reduced renal and cardiac fibrosis in 5/6NX Wistar rats on a high-salt diet, with these effects being comparable to those of angiotensin receptor blocker telmisartan [17]. In addition, Ali et al. [18] demonstrated the beneficial dose-dependent effects of canagliflozin (10 and 25 mg/kg/day) on adenine-induced CKD, demonstrating the improvement of renal parameters (albuminuria, creatinine clearance, and plasma urea) together with the attenuation of inflammation and oxidative stress. We cannot offer a plausible explanation for a lack of any positive effects of the generally accepted dose of empagliflozin in our experimental CKD models, although positive effects were obtained in our previous studies with hypertensive rat strains [22,23]. However, we cannot exclude the possibility that eight weeks following nephrectomy or 2K1C were insufficient for appropriate CKD development, although both of the other researchers and our previous experiments demonstrated a significant increase in proteinuria and a maximal decrease in creatinine clearance after the procedure [29,30,31,32]. Moreover, according to our studies in non-diabetic animals, it seems that the rat models with signs of metabolic disturbances respond to gliflozin therapy more effectively than those without them. This question remains to be analyzed in future studies. 

Apart from the antiproteinuric effects of gliflozins in clinical trials in non-diabetic patients, blood pressure lowering was reported to be only 4-6 mm Hg [5]. Unexpectedly, we did not demonstrate any effect on blood pressure (monitored either by tail plethysmography or by direct BP measurement at the end of the study) in either of the examined models. In contrast, BP was reduced by about 25 mm Hg in Ren-2 transgenic rats [22]. Similar BP reduction was shown by Kim et al. [15] or Rajasekeran et al. [13] in uninephrectomized SD rats, while Zhang et al. [12] or Li et al. [14] did not find any blood pressure effect in the same model. Wan et al. [16] reported the blood-pressure-lowering effects of luseogliflozin measured by telemetry in uninephrectomized Wistar rats kept on high-salt intake. In this context, a comparative study evaluating the same treatment and dose in different rat strains would be rational.

Similar to our [21,22,23] and other investigations [13,19], we have found that relative kidney weight was increased in FHH and UNX-HS rats. Whether this increase is associated with the dilatation of tubular lumen due to SGLT-2-induced diuresis or due to tubular cell hypertrophy remains to be determined. In any case, it is not restricted only to empagliflozin treatment [19,21,22,23], as was also demonstrated with dapagliflozin administration [13].

## 5. Limitations of the Study

There are several limitations of our study concerning the translation of negative results into clinical practice. First of all, our study, similar to other investigators [13,17,20], evaluated the effects of empagliflozin treatment for a relatively short time (8 weeks), which is a substantially shorter duration than that applied in big clinical trials. Thus, our study could be negatively affected by the known effect of SGLT-2 inhibitors on the decreasing glomerular filtration rate and other renal functional parameters in the first weeks of gliflozin treatment. However, other studies performed by our group for the same time of empagliflozin administration [21,22,23] demonstrated the beneficial effects of empagliflozin, relying on different mechanisms of gliflozins in different non-diabetic hypertensive rat strains—metabolic mechanisms in hereditary hypertriglyceridemic rats [21], antihypertensive mechanisms in Ren-2 transgenic [22], or renoprotective mechanisms in spontaneously hypertensive rats expressing C-reactive protein [23]. Second, empagliflozin was the only SGLT-2 inhibitor used, so that the results cannot be simply generalized to all gliflozins. Moreover, it seems that newer gliflozins, such as luseogliflozin, ipragliflozin or ertugliflozin, could have better outcomes than the older ones (dapagliflozin, empagliflozin, and canagliflozin) or the combined SGLT1/2 inhibitors. 

On the other hand, the analysis of Certikova Chabova and Cervenka [33] dealt with the opposite situation—the positive experimental results of dual RAS therapy (combined ACEi and ARB blockade) could not be translated into clinical studies using this drug combination in the treatment of cardiovascular diseases. They discussed not only the role of the variability of disease features in humans versus its relative stability in animals (including the genetic uniformity of animals) but also the selection of animal species for experimental studies. There is no doubt that rats and mice are very different from humans, and therefore pigs or primates would be more appropriate for translational research. However, there are strong limitations concerning animal welfare that almost exclude the performance of such studies.

## 6. Conclusions

In our experimental models of chronic kidney disease, empagliflozin did not provide the expected beneficial effects on kidney function. Whether this lack of kidney protection is due to the short duration of the study, the models used, or the gliflozin administered remains to be determined. However, this finding provides further evidence about the cautious usage of this class of drugs in humans suffering from renal impairment.

## Figures and Tables

**Figure 1 biomedicines-10-02509-f001:**
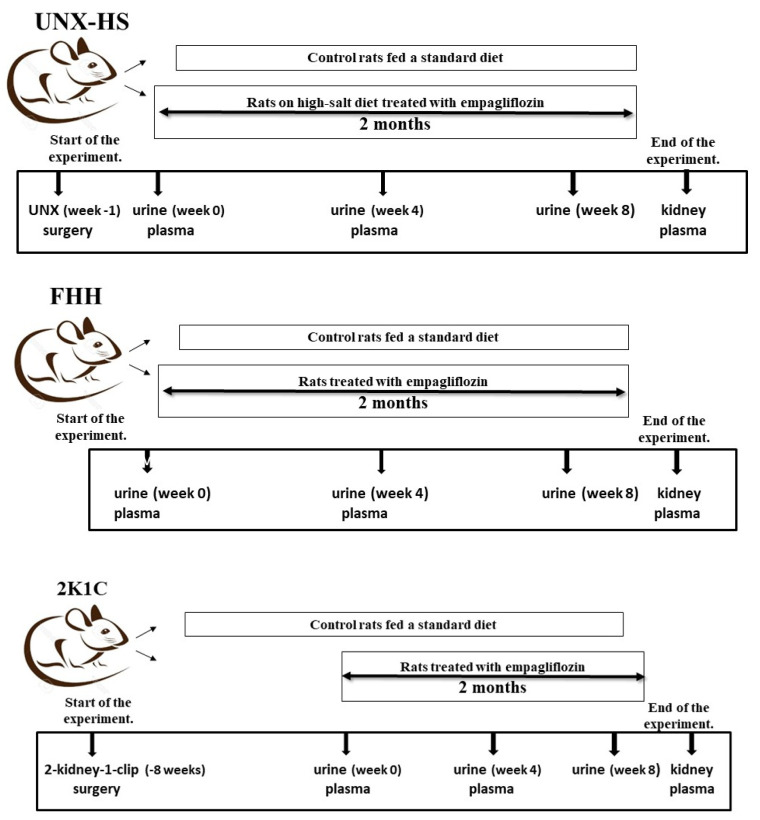
Scheme of the experiment in Fawn-hooded hypertensive rats (FHH), 2-kidney-one-clip (2K1C), and uninephrectomized high salt-fed (UNX-HS) rats.

**Figure 2 biomedicines-10-02509-f002:**
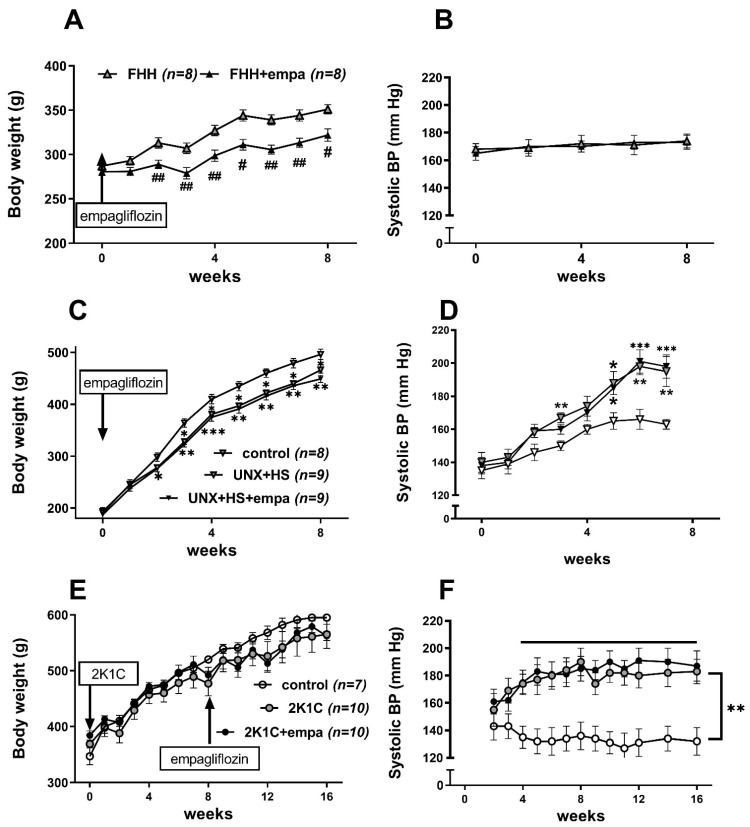
The effect of empagliflozin treatment on body weight (**A**,**C**,**E**) and systolic blood pressure (**B**,**D**,**F**) in fawn-hooded hypertensive (FHH) rats (**A**,**B**), uninephrectomized (UNX) Hannover Sprague Dawley (HanSD) rats on a high-salt (HS) diet (**C**,**D**), and Wistar rats subjected to renal artery stenosis (Goldblatt 2K1C hypertension) (**E**,**F**). * *p* < 0.05 vs. control group, ^#^ *p* < 0.05 vs. untreated group. * denotes *p* < 0.05; ** and ^##^ denotes *p* < 0.01; *** denotes *p* < 0.001; data are means ± SEM. Horizontal line in (**F**) depicts time points with significantly different BP levels.

**Figure 3 biomedicines-10-02509-f003:**
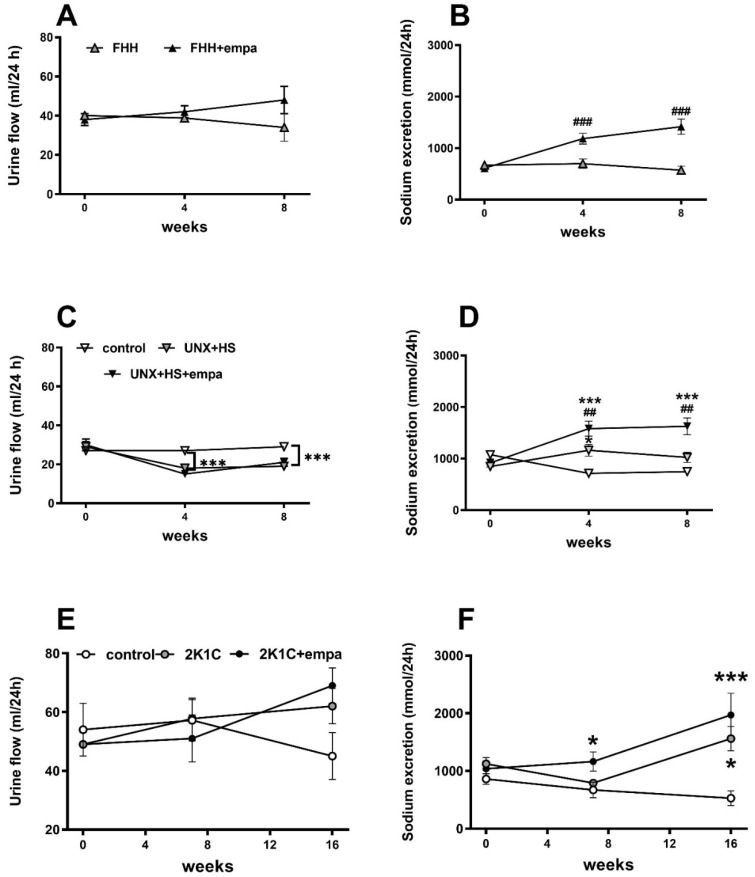
The effect of empagliflozin treatment on diuresis (**A**,**C**,**E**) and sodium excretion (**B**,**D**,**F**) in fawn-hooded hypertensive (FHH) rats (**A**,**B**), uninephrectomized (UNX) Hannover Sprague Dawley (HanSD) rats on high-salt (HS) diet (**C**,**D**), and Wistar rats subjected to renal artery stenosis (Goldblatt 2K1C hypertension) (**E**,**F**). * *p* < 0.05 vs. control group, # *p* < 0.05 vs. untreated group. * denotes *p* < 0.05; ^##^ denotes *p* < 0.01; and ^###^ and *** denotes *p* < 0.001; data are means ± SEM.

**Figure 4 biomedicines-10-02509-f004:**
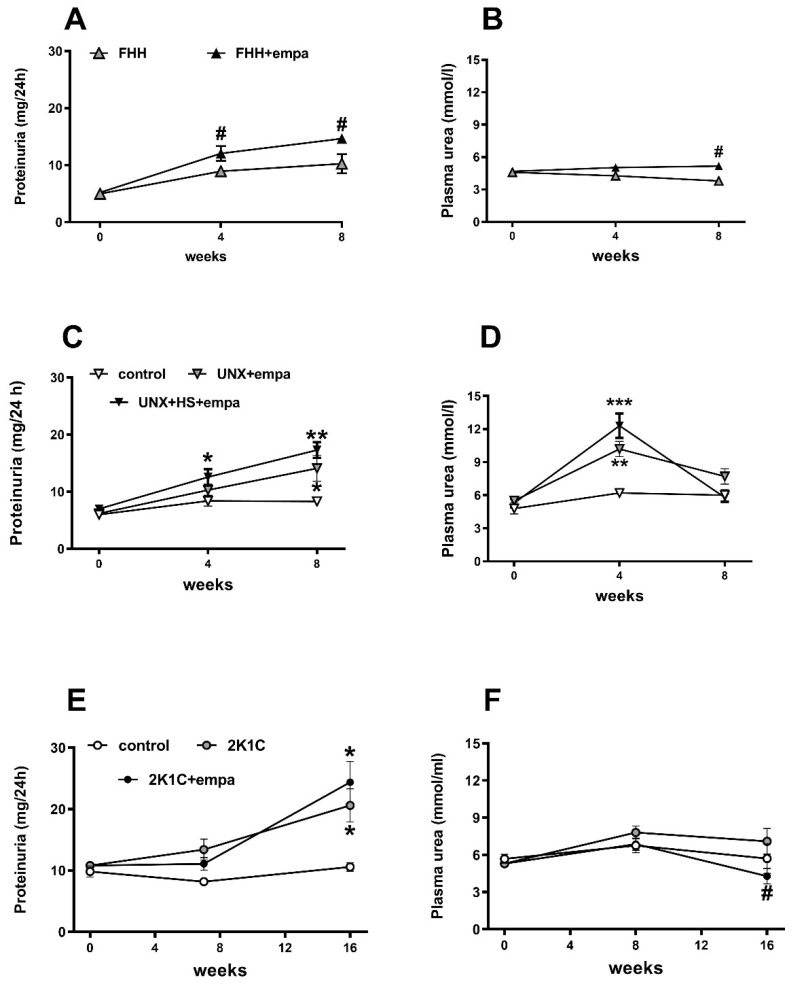
The effect of empagliflozin treatment on proteinuria (**A**,**C**,**E**) and plasma urea (**B**,**D**,**F**) in fawn-hooded hypertensive (FHH) rats (**A**,**B**), uninephrectomized (UNX) Hannover Sprague Dawley (HanSD) rats on high-salt (HS) diet (**C**,**D**), and Wistar rats subjected to renal artery stenosis (Goldblatt 2K1C hypertension) (**E**,**F**). * *p* < 0.05 vs. control group, # *p* < 0.05 vs. untreated group. * denotes *p* < 0.05; ** denotes *p* < 0.01; and *** denotes *p* < 0.001; data are means ± SEM.

**Figure 5 biomedicines-10-02509-f005:**
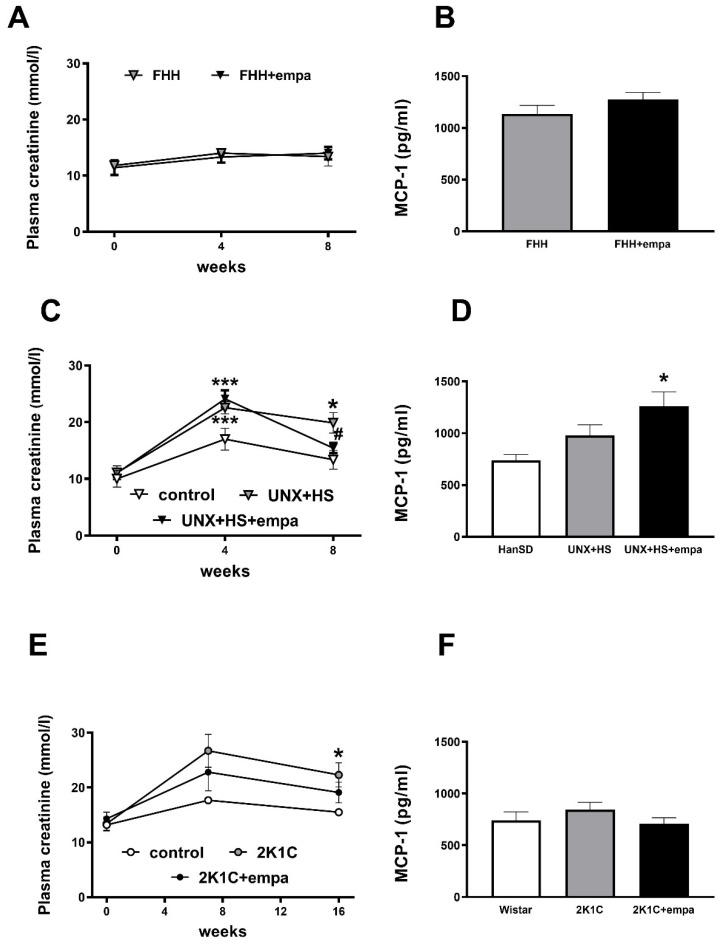
The effect of empagliflozin treatment on plasma creatinine (**A**,**C**,**E**) and on plasma monocyte attractant protein-1 (MCP-1) (**B**,**D**,**F**) in fawn-hooded hypertensive rats (**A**,**B**), uninephrectomized (UNX) Hannover Sprague Dawley (HanSD rats on high-salt (HS) diet (**C**,**D**), and Wistar rats subjected to renal artery stenosis (Goldblatt 2K1C hypertension) (**E**,**F**). * *p* < 0.05 vs. control group, # *p* < 0.05 vs. untreated group. * denotes *p* < 0.05 and *** denotes *p* < 0.001; data are means ± SEM.

**Table 1 biomedicines-10-02509-t001:** Body and organ weights, as well as blood pressure and TBARS levels, in fawn-hooded hypertensive (FHH) rats treated with empagliflozin (empa).

	FHH Untreated	FHH + empa
Body weight (g)	351 ± 5	322 ± 7 ^#^
Relative heart weight (g/100 g BW)	0.287 ± 0.007	0.286 ± 0.007
Relative kidneys weight (g/100 g BW)	0.767 ± 0.028	0.900 ± 0.036 ^#^
Relative weight of epididymal fat (g/100 g BW)	0.724 ± 0.016	0.603 ± 0.025 ^#^
Relative weight of perirenal fat (g/100 g BW)	0.289 ± 0.018	0.176 ± 0.018 ^#^
Systolic BP (mm Hg)	180 ± 7	173 ± 5
TBARS in kidneys	36.8 ± 1.4	39.3 ± 1.3

* denotes *p* < 0.05 vs. control group; **^#^** denotes *p* < 0.05 vs. untreated group; data are means ± SEM; and *n* = 7–10 for each group.

**Table 2 biomedicines-10-02509-t002:** Body and organ weights as well as blood pressure and TBARS levels in uninephrectomized salt-loaded (UNX + HS) Hannover Sprague Dawley (HanSD) rats treated with empagliflozin (empa).

	HanSD Control	UNX + HS	UNX + HS + empa
Body weight (g)	472 ± 10	461 ± 6	447 ± 7
Relative heart weight (g/100 g BW)	0.27 ± 0.02	0.33 ± 0.02 *	0.32 ± 0.02 *
Relative left ventricle (g/100g)	0.212 ± 0.005	0.264 ± 0.007 *	0.261 ± 0.009 *
Relative left kidney weight (g/100 g BW)	0.37 ± 0.01	0.74 ± 0.94 *	0.94 ± 0.04 *^,#^
Relative epididymal fat (g/100 g BW)	1.356 ± 0.081	0.891 ± 0.056 *	0.812 ± 0.028 *
Relative retroperitoneal fat (g/100 g BW)	1.349 ± 0.076	0.795 ± 0.077 *	0.678 ± 0.051 *
Systolic BP (mm Hg)	127 ± 57	166 ± 8 *	179 ± 8 *
TBARS in kidney	33 ± 3	44 ± 4 *	45 ± 2 *

* denotes *p* < 0.05 vs. control group; **^#^** denotes *p* < 0.05 vs. untreated group; data are means ± SEM; and *n* = 7–10 for each group.

**Table 3 biomedicines-10-02509-t003:** Body and organ weights, as well as blood pressure and TBARS levels, in Wistar rats subjected to renal artery stenosis (Goldblatt 2K1C hypertension) treated with empagliflozin (empa).

	Wistar Control	2K1C	2K1C + empa
Body weight (g)	595 ± 12	565 ± 26	562 ± 13
Relative heart weight (g/100 g BW)	0.212 ± 0.004	0.269 ± 0.017 *	0.269 ± 0.011 *
Relative left ventricle weight (g/100 g BW)	0.173 ± 0.003	0.226 ± 0.017 *	0.223 ± 0.010 *
Relative left kidney weight (g/100 g BW)	0.303 ± 0.006	0.180 ± 0.032 *	0.220 ± 0.030 *
Relative right kidney weight (g/100 g BW)	0.307 ± 0.007	0.419 ± 0.022 *	0.491 ± 0.028 *
Relative epididymal fat (g/100 g BW)	1.47 ± 0.06	1.33 ± 0.17	1.33 ± 0.13
Relative perirenal fat (g/100 g BW)	1.29 ± 0.10	1.15 ± 0.24	1.20 ± 0.17
Systolic BP (mm Hg)	120 ± 4	163 ± 9 *	157 ± 10 *
TBARS in kidney	31 ± 3	20 ± 2	24 ± 3

* denotes *p* < 0.05 vs. control group; **^#^** denotes *p* < 0.05 vs. untreated group; data are means ± SEM; and *n* = 7–10 for each group.

## Data Availability

All data arising from this study are contained within the article.

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
