# Peer review of "Empagliflozin Is Not Renoprotective in Non-Diabetic Rat Models of Chronic Kidney Disease"

_biomedicines, 2022, doi:10.3390/biomedicines10102509_

Round 1

Reviewer 1 Report

In the present manuscript, Hojna et al., demonstrate the effect Empagliflozin in three models of

 chronic kidney disease (CKD): Fawn-hooded hypertensive (FHH) rats, 15 uninephrectomized salt-loaded (UNX+HS) rats and in rats with Goldblatt hypertension. Authors treated these mice with Empagliflozin and evaluated various biochemical parameters in blood, kidney and liver. Authors claim that Empagliflozin did not provide any expected beneficial effects on kidney function in 3 tested CKD experimental models based on the parameters analyzed. The manuscript is potentially interesting and asks relevant question. The experiments are well designed but the presentation of results and the text in the manuscript is poor with several errors and needs attention. Moreover, I have following comments for the authors to take into consideration.

1)    The main observations of the manuscript are related with the Empagliflozin treatment and experimental schemes of Empagliflozin treatment. But in the manuscript, these are not clearly stated by the authors. I would suggest diagrams representation that provide clear description for the experimental schemes/ strategy for each CKD model with treatment plan, time points of diet start/stop, samples collected for analysis etc.

2)    Authors state that Empagliflozin was given in diet. Did authors check for weekly diet consumption? And how did it vary from control diet?

3)    Authors should analyze plasma concentration of protein-bound and unbound Empagliflozin after treatment.

4)    Why only male rats were used for the study? Were there any sex related differences?

5)    What does the line in figure 1F denote? Also, in figure 1, there is no error bar in some places. Did all the rats behaved similarly or it’s the representative line graph?

6)    In figure 2 A and B, which symbol represent which treatment group?

7)    In figure 3 in some graphs x-axis is labelled in others its not. Y-axis legends is of different size in each graph. Please be consistent with figure representation as it becomes confusing to understand.

8)    In figure 4, At what time point MCP-1 was analyzed and why only one time-point was used unlike other parameters where different time points were used?

9)    Authors should discuss the possible reason why Empagliflozin increased kidney weight in all of the studied model?

10) Authors should analyze Kidney histology for fibrosis using standard Masson-Goldner staining. Also, it would be worthwhile to examine mRNA abundance of fibronectin, alpha-SMA and collagen1a1 by qPCR.

11) Authors need to thoroughly revise the manuscript text and result presentation.

Author Response

1)           The main observations of the manuscript are related with the Empagliflozin treatment and experimental schemes of Empagliflozin treatment. But in the manuscript, these are not clearly stated by the authors. I would suggest diagrams representation that provide clear description for the experimental schemes/ strategy for each CKD model with treatment plan, time points of diet start/stop, samples collected for analysis etc.

Response to the reviewer:

Diagrams showing the experimental scheme of the experiment for all three CKD models were added to the manuscript.

2)           Authors state that Empagliflozin was given in diet. Did authors check for weekly diet consumption? And how did it vary from control diet?

Response to the reviewer:

The authors thank for this question. Yes, we measured the food consumption during the study. The results on food consumption were added to the manuscript (lines 138-140).   

3)    Authors should analyze plasma concentration of protein-bound and unbound Empagliflozin after treatment.

Response to the reviewer:

Unfortunately, we are not able to measure the plasma concentrations of empagliflozin since plasma samples for further analyses are not available.

4)    Why only male rats were used for the study? Were there any sex related differences?

Response to the reviewer:

This is a very interesting question. This study was not primarily designed to analyze gender differences in the sensitivity to SGLT-2 inhibitors. Moreover, since we wanted to compare our present results with our previous studies performed in male rats, we used only male rats. The main reason for not using female rats in our studies is the estrus cycle which can interfere with many physiological systems.

5)    What does the line in figure 1F denote? Also, in figure 1, there is no error bar in some places. Did all the rats behaved similarly or it’s the representative line graph?

Response to the reviewer:

The line from Figure 1F depicts time points at which significantly higher BP was found.

All figures plot the average values+SEM for each group (n=7-10 animals). Unfortunately some SE error lines were too small to be visible. Therefore, we changed the scale in Fig. 1A and 1C and also the size of symbols in Figure 1C to overcome this problem.

6)    In figure 2 A and B, which symbol represent which treatment group?

Response to the reviewer:

Sorry for our mistake, the symbols were corrected to be visible.

7)    In figure 3 in some graphs x-axis is labelled in others is not. Y-axis legends is of different size in each graph. Please be consistent with figure representation as it becomes confusing to understand.

Response to the reviewer:

We have corrected x and y axis according to the reviewer comments.

8)    In figure 4, At what time point MCP-1 was analyzed and why only one time-point was used unlike other parameters where different time points were used?

Response to the reviewer:

Unlike other functional renal parameters, which were evaluated as successive changes during the study, MCP-1 levels were detected only at the end of the study. Their levels are recognized as marker of advanced CKD (especially of fibrosis and tubular damage) (Bullen et al, Am J Kidney Dis, 2021). We appreciate this comment very much and will include it in our future experiments.

9)    Authors should discuss the possible reason why Empagliflozin increased kidney weight in all of the studied model?

Response to the reviewer:

The discussion on kidney weight was included to the Discussion (lines 285-290).

“Similar to our [21-23] and other investigations [13, 19], we have found that relative kidney weight was increased in FHH and UNX-HS rats. Whether this increase is associated with the dilatation of tubular lumen due to SGLT-2-induced diuresis or due to tubular cell hypertrophy remains to be determined. In any case, this phenomenon is not restricted only to empagliflozin treatment [19, 21-23] as it was also demonstrated with dapagliflozin administration [13].”

10) Authors should analyze Kidney histology for fibrosis using standard Masson-Goldner staining. Also, it would be worthwhile to examine mRNA abundance of fibronectin, alpha-SMA and collagen1a1 by qPCR.

Response to the reviewer:

Due to the fact that none of our previous studies in non-diabetic hypertensive models (Huttl et al, 2019; Hojná et al, 2021; Malinska et al, 2022) has demonstrated any effect of empagliflozin on glomerular or tubular damage (evaluated as glomerulosclerosis index and tubulointerstitial injury) despite its beneficial effect on kidney function, we did not perform the histological analysis, especially in respect to rather negative results of the present study. If the beneficial effects are present, we typically analyze TGF-β, TNF-α, collagen and fibronectin in kidney samples by RT-PCR. 

11) Authors need to thoroughly revise the manuscript text and result presentation.

Response to the reviewer:

We highly appreciate all comments of the reviewer which helped us to improve the manuscript. We have tried our best to answer all the raised questions.

Reviewer 2 Report

Major point

Results from animal studies do not necessarily match those from clinical trials. However, in order to prove the effectiveness of clinical trials, the current research process would be to elucidate the mechanism under similar conclusions. Although it cannot be said that all of the negative results are failed studies, some degree of correlation with the results of clinical studies is required for the negative results to be meaningful. Although this study may give information that the results and clinical results are not consistent with the experimental results on animals designed by the researcher, it is judged that it is too difficult to publish this as a article. In addition, I think that it is a natural result to say that the mechanism also does not match in a situation where the results do not match. First of all, it will be necessary to design a new animal so that the animal experiment itself can be consistent with the clinical results to some extent.

Author Response

Responses to the Reviewer 2

Results from animal studies do not necessarily match those from clinical trials. However, in order to prove the effectiveness of clinical trials, the current research process would be to elucidate the mechanism under similar conclusions. Although it cannot be said that all of the negative results are failed studies, some degree of correlation with the results of clinical studies is required for the negative results to be meaningful. Although this study may give information that the results and clinical results are not consistent with the experimental results on animals designed by the researcher, it is judged that it is too difficult to publish this as a article. In addition, I think that it is a natural result to say that the mechanism also does not match in a situation where the results do not match. First of all, it will be necessary to design a new animal so that the animal experiment itself can be consistent with the clinical results to some extent.

Response to the Reviewer

We highly appreciate the comment of the reviewer concerning the translation of experimental results into clinical practice and vice versa. We are aware of these limitations and therefore we included the paragraph dealing with this topic at the end of the manuscript.

Limitations of the study

There are several limitations of our study concerning the translation of negative results into clinical practice. First of all, our study, similar to other investigators [13, 17, 20] evaluated the effects of empagliflozin treatment for a relatively short time (8 weeks), which is substantially shorter duration than that applied in big clinical trials. Thus, our study could be negatively affected by the known effect of SGLT-2 inhibitors on decreasing glomerular filtration rate and other renal functional parameters in the first weeks of gliflozin treatment. However, other studies performed by our group for the same time of empagliflozin administration [21-23] demonstrated beneficial effects of empagliflozin relying in different mechanisms of gliflozins in three non-diabetic hypertensive rat strains – metabolic mechanisms in hereditary hypertriglyceridemic rats [21], antihypertensive mechanisms in Ren-2 transgenic rats[22], or renoprotective mechanisms in spontaneously hypertensive rats expressing C-reactive protein [23]. Second, empagliflozin was the only SGLT-2 inhibitor used, so that the results cannot be simply generalized to all gliflozins. Moreover, it seems, that newer gliflozins, such as luseogliflozin, ipragliflozin or ertugliflozin, could have better outcomes than the older ones (dapagliflozin, empagliflozin, canagliflozin) or the combined SGLT1/2 inhibitors.

On the other hand, the analysis of Certikova Chabova and Cervenka [33] dealt with the opposite situation - the positive experimental results of dual RAS therapy (combined ACEi and ARB blockade) could not be successfully translated into clinical studies using this drug combination in the treatment of cardiovascular diseases. They discussed not only the role of variability of disease features in humans versus its relative stability in animals (including the genetic uniformity of animals) but also a selection of animal species for experimental studies. There is no doubt that rats and mice are far from humans and therefore rather pigs or primates would be appropriate for translational research. However, there are strong limitations concerning animal welfare that almost exclude performing of such studies.

Reviewer 3 Report

The manuscript by Hojná Set al entitled "Empagliflozin is not renoprotective in non-diabetic rat models of chronic kidney disease" is lacks novelty and poorly written. There are the following issues that need to be fixed before consideration for publication.

1. Authors should have used a positive control drug to compare the effects of Empagliflozin.

2. The authors mentioned that food intake was measured but did not provide the data. Please provide the food intake data as Empagliflozin is mixed into the diet and may affect the observed findings if food intake is not the same among the groups. This will also demonstrate the amount of Empagliflozin intake. Ideally, Empagliflozin should have been administered orally to avoid variation between animals due to differences involuntery eating.

3. Authors need to be more clear and affirmative when writing about the findings. eg Authors wrote, "Relative weights of organs and visceral fat were substantially affected, whereas empagliflozin had no effect in both experimentally induced CKD models". It does not make clear sense if the treatment increased body weight or decreased it. Also please provide a relative change in body weight when comparing between two groups.

4. The data presented in tables should be represented in figures as scatter dot plots to represent data from all the animals and variability among the group.

5. In fig 1B and 1F, please correct units of Systolic BP.

6. Authors wrote in the statistics section that data were analysed by either two way ANOVA or one way ANOVA, which does not clarify which data were analysed by which test. Please specify the statistics used for each data in the figure legends. 

Author Response

Responses to the Reviewer 3

The manuscript by Hojná Set al entitled "Empagliflozin is not renoprotective in non-diabetic rat models of chronic kidney disease" is lacks novelty and poorly written. There are the following issues that need to be fixed before consideration for publication.

  1. Authors should have used a positive control drug to compare the effects of Empagliflozin.

Response to the Reviewer

We highly appreciate the comment on using a positive control drug for comparison of empagliflozin effects. Such study has already been performed by our group as the extension of our previous study (Hojná et al, 2021), in which beneficial effects of empagliflozin treatment were demonstrated in non-diabetic hypertensive Ren-2 transgenic rats. We have used a low dose of ACE inhibitor trandolapril and compared it with a triple therapy (hydralazine, hydrochlorothiazide and reserpine) in order to see the additional effects of combined therapy with empagliflozin. These results are being prepared for publication.

  1. The authors mentioned that food intake was measured but did not provide the data. Please provide the food intake data as Empagliflozin is mixed into the diet and may affect the observed findings if food intake is not the same among the groups. This will also demonstrate the amount of Empagliflozin intake. Ideally, Empagliflozin should have been administered orally to avoid variation between animals due to differences involuntery eating.

Responses to the Reviewer

The data (average food intake per day per 100 g of body weight) were included to the result section (lines 138-140). These data show that the food intake did not differ between the untreated and treated animals.

Empagliflozin was mixed to the diet in the amount, which was based on our previous studies.

According to our experience, in the chronic studies we prefer drug administration through the diet since oral administration by gavage is a major stress to the animals.

  1. Authors need to be more clear and affirmative when writing about the findings. eg Authors wrote, "Relative weights of organs and visceral fat were substantially affected, whereas empagliflozin had no effect in both experimentally induced CKD models". It does not make clear sense if the treatment increased body weight or decreased it. Also please provide a relative change in body weight when comparing between two groups.

Response to the Reviewer

The authors thank the reviewer for this comment. We have tried to describe the body parameters in more detail (lines 141-145). The text is now as follows:

“There was no change in blood pressure (measured by tail-cuff plethysmography) in FHH rats during the experiment (Tables 1-3). Moreover, empagliflozin treatment did not modify blood pressure and relative heart weight in this strain. In contrast, blood pressure substantially increased (by about 60 mm Hg in HanSD-UNX+HS and 40 mm Hg in Wistar-2K1C) following surgical procedures in both experimentally induced CKD models (Fig. 1 B, D, F). This was followed by an increase in their relative heart weights. Empagliflozin had no effect on tail-cuff BP in either experimental group. This was also confirmed by a direct BP measurement at the end of the study. Visceral adi-posity was decreased by empagliflozin treatment only in FHH rats. Relative kidney weight was increased in empagliflozn-treated FHH and in uninephrectomized salt-loaded rats while it was unchanged in 2-kidney-1-clip rats.”

Concerning the relative changes in body weights, we tried to depict them in graphs but due to the similar basal body weights, they looked the same. We hope that there is no problem in showing the absolute values than their relative changes.

  1. The data presented in tables should be represented in figures as scatter dot plots to represent data from all the animals and variability among the group.

Response to the Reviewer

We understand the request of the reviewer for scatter plots of values included in the tables, as graphs always show the results more clearly. However, due to 24 parameters in Tables 1-3 we would need at least four more Figures (each with 6 parameters), which is unfortunately not acceptable by the Journal.

  1. In fig 1B and 1F, please correct units of Systolic BP.

Response to the Reviewer

Units were corrected in Figure 1B and F, sorry for our mistake.

  1. Authors wrote in the statistics section that data were analysed by either two way ANOVA or one way ANOVA, which does not clarify which data were analysed by which test. Please specify the statistics used for each data in the figure legends. 

Response to the Reviewer

Data were analyzed by one way ANOVA with Bonferoni test, the text was corrected adequately (lines 127-129).

Round 2

Reviewer 1 Report

The authors have addressed majority of my comments except plasma concentration of protein-bound and unbound Empagliflozin after treatment to analyze how much Empagliflozin was actually take up by the system when fed with diet. I have no further concerns/comments.

Reviewer 2 Report

I think that the authors have adequately corrected their paper according to the reviewer’s recommendations overall. I agree with the limitations of the revision from the authors.

Reviewer 3 Report

The authors have significantly improved the manuscript and answered all my questions/concerns.